# Novel Multimodal Salicylamide Derivative with Antidepressant-like, Anxiolytic-like, Antipsychotic-like, and Anti-Amnesic Activity in Mice

**DOI:** 10.3390/ph16020175

**Published:** 2023-01-24

**Authors:** Elżbieta Żmudzka, Klaudia Lustyk, Monika Głuch-Lutwin, Małgorzata Wolak, Jolanta Jaśkowska, Marcin Kołaczkowski, Jacek Sapa, Karolina Pytka

**Affiliations:** 1Department of Social Pharmacy, Faculty of Pharmacy, Jagiellonian University Medical College, Medyczna 9, 30-688 Krakow, Poland; 2Department of Pharmacodynamics, Faculty of Pharmacy, Jagiellonian University Medical College, Medyczna 9, 30-688 Krakow, Poland; 3Department of Pharmacobiology, Faculty of Pharmacy, Jagiellonian University Medical College, Medyczna 9, 30-688 Krakow, Poland; 4Department of Organic Chemistry and Technology, Faculty of Chemical and Engineering and Technology, Cracow University of Technology, Warszawska 24, 31-155 Krakow, Poland; 5Department of Medicinal Chemistry, Faculty of Pharmacy, Jagiellonian University Medical College, Medyczna 9, 30-688 Krakow, Poland

**Keywords:** antidepressant-like activity, anxiolytic-like action, antipsychotic-like properties, anti-amnesic effect, memory-enhancing, serotonin receptor antagonist

## Abstract

Depression, anxiety, and schizophrenia may coexist in psychiatric patients. Moreover, these disorders are very often associated with cognitive impairments. However, pharmacotherapy of these conditions remains challenging due to limited drug effectiveness or numerous side effects. Therefore, there is an urgent need to develop novel multimodal compounds that can be used to treat depression, anxiety, and schizophrenia, as well as memory deficits. Thus, this study aimed to evaluate the potential antidepressant-like, anxiolytic-like, antipsychotic-like effects, and anti-amnesic properties, of the novel arylpiperazine derivative of salicylamide, JJGW07, with an affinity towards serotonin 5-HT_1A_, 5-HT_2A_, and 5-HT_7_ and dopamine D_2_ receptors. Firstly, we investigated the compound’s affinity for 5-HT_6_ receptors and its functional activity by using in vitro assays. JJGW07 did not bind to 5-HT_6_ receptors and showed antagonistic properties for 5-HT_1A_, 5-HT_2A_, 5-HT_7_, and D_2_ receptors. Based on the receptor profile, we performed behavioral studies in mice to evaluate the antidepressant-like, anxiolytic-like, and antipsychotic-like activity of the tested compound using forced swim and tail suspension tests; four-plate, marble-burying, and elevated plus maze tests; and MK-801- and amphetamine-induced hyperlocomotion tests, respectively. JJGW07 revealed antidepressant-like properties in the tail suspension test, anxiolytic-like effects in the four-plate and marble-burying tests, and antipsychotic-like activity in the MK-801-induced hyperlocomotion test. Importantly, the tested compound did not induce catalepsy and motor impairments or influence locomotor activity in rodents. Finally, to assess the potential procognitive and anti-amnesic properties of JJGW07, we used passive avoidance and object recognition tests in mice. JJGW07 demonstrated positive effects on long-term emotional memory and also ameliorated MK-801-induced emotional memory impairments in mice, but showed no procognitive properties in the case of recognition memory. Our results encourage the search for new compounds among salicylamide derivatives, which could be model structures with multitarget mechanisms of action that could be used in psychiatric disorder therapy.

## 1. Introduction

Depression, anxiety, and schizophrenia may coexist in psychiatric patients. Moreover, these disorders are very often associated with cognitive impairments. Untreated or badly-diagnosed mental illnesses lead to a decreased quality of life, difficulties with social functioning, alcohol or drug addictions, and a higher risk of suicide [1]. Unfortunately, the prevalence of mental health problems and cognitive deficits has increased significantly in the past few years [2]. The World Health Organization reported that one in every eight people suffers from a mental disorder worldwide [2]. Additionally, the COVID-19 pandemic has significantly worsened mental health by forcing lockdowns, social distancing, and community restrictions [3]. Sadly, affected patients struggle not only with functional disabilities or emotional dysregulation, but also with memory deficits [4].

Unfortunately, despite numerous antipsychotics, antidepressants, and anxiolytics on the market, many patients do not respond to treatment. Furthermore, the available therapeutic options have many side effects, which are often a reason for drug discontinuation. Therefore, considering the limited effectiveness and safety of the available therapeutics, searching for novel drugs that improve the pharmacotherapy of mental disorders is one of the leading health challenges of the 21st century.

Thus, in our search for centrally-acting compounds, we investigated arylpiperazine derivatives of salicylamide with an affinity for serotonin and dopamine receptors [5]. We have previously shown the antipsychotic-like and anxiolytic-like effects [6], as well as the memory-enhancing properties (manuscript under review), of JJGW08. In this study, we selected another arylpiperazine derivative of salicylamide, JJGW07, due to its high affinity for serotonin 5-HT_1A_, moderate affinity for 5-HT_7_ and D_2_, and weak affinity for 5-HT_2A_ receptors [5]. As these receptors are widely distributed in various brain regions, they are associated with diverse CNS disorders, including depression, anxiety, schizophrenia, and memory disturbances [7,8,9,10,11,12,13]. Scientists have discovered that both agonists and antagonists of the 5-HT_1A_ receptor may possess antidepressant and/or anxiolytic activity [9], whereas agonists or partial agonists may demonstrate antipsychotic or procognitive potential [14,15,16,17]. Considering serotonin 5-HT_2A_ receptors, their blockade may improve the symptoms of depression, anxiety, and schizophrenia [9,18]. Further, serotonin 5-HT_7_ receptor antagonists may possess antidepressant and antipsychotic activity, whereas both agonists and antagonists can demonstrate anxiolytic effects and memory-enhancing properties [9,14,19,20,21,22]. Finally, dopamine D_2_ receptor antagonists or partial agonists may reduce the symptoms of schizophrenia, whereas agonists can ameliorate depression-like conditions [8,23,24,25]. Taking the above into account, as well as considering the receptor profile of JJGW07 and our previous studies on the arylpiperazine derivative of salicylamide, we aimed to assess the antidepressant-like, anxiolytic-like, and antipsychotic-like effects of JJGW07 in mice. Moreover, we determined the type of interaction of JJGW07 with the 5-HT_1A_, 5-HT_2A_, 5-HT_7,_ and D_2_ receptors and initially evaluated its central nervous system safety. Finally, we assessed the potential procognitive and anti-amnesic properties of JJGW07 in mice.

## 2. Results

### 2.1. JJGW07 Showed No Affinity for 5-HT_6_ Receptors

JJGW07 possessed no affinity for 5-HT_6_ receptors and did not bind to the receptors at a concentration of 10^−5^ M, whereas the *p*Ki value for methiotepine, a reference compound, was 8.48 ± 0.04.

### 2.2. JJGW07 Showed Antagonistic Properties at D_2_, 5-HT_1A_, 5-HT_2A_, and 5-HT_7_ Receptors

In functional assays, the tested compound demonstrated strong antagonistic properties at dopamine D_2_ and serotonin 5-HT_1A_ receptors and weak antagonistic properties towards serotonin 5-HT_2A_ and 5-HT_7_ receptors (Table 1).

### 2.3. JJGW07 Did Not Affect Mouse Immobility in the Forced Swim Test and Reduced the Immobility Time in the Tail Suspension Test

JJGW07 affected neither the immobility time (F(3,30) = 2.466, *p* = 0.081) nor the swimming time (H(4,32) = 9.403, *p* = 0.024) or the climbing time (F(4,32) = 2.529, *p* = 0.097) in the forced swim test in mice (Figure 1A).

On the other hand, JJGW07 reduced the immobility time in the tail suspension test at the doses of 0.15, 0.3, and 0.625 mg/kg by 50, 41, and 39% (F(6,63) = 7.178, *p* < 0.0001), respectively (Figure 1B).

### 2.4. JJGW07 Increased the Number of Punished Crossings in the Four Plate Test in Mice

JJGW07 increased the number of punished crossings of animals by 36% at a dose of 1.25 mg/kg (F(3,28) = 4.859, *p* = 0.008) (Figure 2A).

### 2.5. JJGW07 Decreased the Number of Buried Marbles in the Marble-Burying Test in Mice

JJGW07 significantly reduced the number of buried marbles at the doses of 0.625 and 1.25 mg/kg by 80 and 66%, respectively (F(4,39) = 5.749, *p* = 0.001) (Figure 2B).

### 2.6. JJGW07 Did Not Increase the Time Spent in Open Arms in the Elevated Plus Maze Test in Mice

JJGW07 did not significantly affect the time spent in open arms of the apparatus (F(3,28) = 3.314, *p* = 0.034), but the tested doses significantly reduced the number of entries into open arms by 54, 43, and 49% (F(3,28) = 4.036, *p* < 0.05) (Figure 2C,D).

### 2.7. JJGW07 Did Not Influence the Locomotor Activity in Mice

JJGW07 did not affect the locomotor activity in mice in the 60 min session (F(4,32) = 2.269, *p* = 0.111), 30 min session (F(4,35) = 2.407, *p* = 0.068), 6 min session (F(6,51) = 2.218, *p* = 0.056), 5 min session (H(4,36) = 1.417, *p* = 0.702), 4 min session (F(3,32) = 0.625, *p* = 0.603), or 1 min session (F(3,31) = 0.408, *p* = 0.748) (Table 2).

### 2.8. JJGW07 Did Not Influence the Motor Coordination in Mice

JJGW07 did not affect the motor coordination in mice. The number of animals that fell from the rotating rod, the time before animals fell, and the TD_50_ values are presented in Table 3.

### 2.9. JJGW07 Reversed the MK-801- and Amphetamine-Induced Hyperlocomotion in Mice

The tested compound reduced the hyperactivity of animals induced by MK-801 at the doses of 0.625, 1.25, and 2.5 mg/kg by 40, 51, and 50%, respectively (F(6,51) = 10.17, *p* < 0.0001) (Figure 3A). On the other hand, JJGW07 did not decrease the amphetamine-induced hyperlocomotion in mice at the dose range of 0.625–2.5 mg/kg (F(5,44) = 7.261, *p* = 0.0006) (Figure 3B).

### 2.10. JJGW07 Did Not Induce Catalepsy in the Bar Test in Mice at Antipsychotic-like Doses

JJGW07 induced catalepsy at a dose of 10 mg/kg (Table 4).

### 2.11. JJGW07 Did Not Disturb Long-Term Memory in Naïve Mice in the Step-through Passive Avoidance Task

JJGW07 did not influence the latency time in the acquisition trial, whereas it significantly increased the latency time in the retention trial at all antipsychotic-like doses in comparison to the acquisition session. The statistical analysis showed a significant time effect (F(1,28) = 46.04, *p* < 0.0001), but no influence of the compound (F(3,28) = 0.681, *p* = 0.571) and no interaction (F(3,28) = 0.677, *p* = 0.573) (Figure 4A).

Aripiprazole, used as a reference compound at the doses of 0.25–2.0 mg/kg, did not influence the latency time in the acquisition trial, whereas it significantly increased the latency time in the retention trial at all tested doses compared to the acquisition session. The statistical analysis showed a significant time effect (F(1,35) = 91.76, *p* < 0.0001), but no influence of the compound (F(4,35) = 1.627, *p* = 0.189) and no interaction (F(4,35) = 1.538, *p* = 0.213) (Figure 4B).

### 2.12. JJGW07 Reversed Cognitive Disturbances after the MK-801 Administration in Mice in the Step-Through Passive Avoidance Task

JJGW07 or MK-801 did not influence the latency in the acquisition trial. JJGW07 reversed the MK-801-induced memory impairment at a dose of 1.25 mg/kg, increasing the latency time in the retention session. The statistical analysis showed a significant time effect (F(1,35) = 44.46, *p* < 0.0001), a significant influence of the compound (F(4,35) = 3.531, *p* = 0.016), and an interaction (F(4,35) = 3.381, *p* = 0.019) (Figure 5A).

Aripiprazole at the doses of 0.25–2.0 mg/kg did not influence the latency time in the acquisition trial or the retention session. The statistical analysis showed a significant time effect (F(1,42) = 34.47, *p* < 0.0001), a significant influence of the compound (F(5,42) = 3.353, *p* = 0.012), and an interaction (F(5,42) = 4.204, *p* = 0.003) (Figure 5B).

### 2.13. JJGW07 Did Not Disturb Long-Term Memory in Naïve Mice in the Object Recognition Test

JJGW07 did not increase the exploration of a novel object significantly higher than the chance level of 10 s, which may suggest its impact on memory retention (Figure 6A). On the other hand, all the tested doses of aripiprazole were significantly different from the chance level, indicating that the reference compound did not interfere with the increased exploration of the novel object (Figure 6B). Moreover, aripiprazole at the doses of 0.25 and 0.5 mg/kg increased the exploration level of the novel object significantly higher than the chance level of 10 s, reversing the effect of MK-801 (Figure 6C).

## 3. Discussion

In this study, we evaluated the antidepressant-like, anxiolytic-like, and antipsychotic-like effects as well as the anti-amnesic properties of a novel salicylamide derivative, JJGW07, which, in a previous study, showed affinity towards serotonin 5-HT_1A_, 5-HT_2A_, and 5-HT_7_ and dopamine D_2_ receptors [5]. JJGW07 showed no affinity for 5-HT_6_ receptors in the radioligand binding assay. In functional assays, the compound acted as an antagonist at 5-HT_1A_, 5-HT_2A_, 5-HT_7_, and D_2_ receptors. Furthermore, JJGW07 revealed antidepressant-like properties in the tail suspension test, but not in the forced swim test in mice, as well as anxiolytic-like effects in the four-plate and marble-burying tests, but not in the elevated plus maze test, in mice. Moreover, JJGW07 presented antipsychotic-like activity, reducing the hyperactivity of mice induced by MK-801, but not by amphetamine. Notably, the tested compound did not impair the motor coordination and did not induce catalepsy in mice at active doses. What is more, JJGW07 showed procognitive and anti-amnesic properties in the passive avoidance task, but not in the object recognition test in mice.

Previous studies have demonstrated that JJGW07 had an affinity for serotonin 5-HT_1A_, 5-HT_2A_, and 5-HT_7_ and dopamine D_2_ receptors [5]. Such multitarget-directed ligands might show the multidirectional pharmacological effects in the CNS and thus may have potential applications in treating many neuropsychiatric diseases. Therefore, as a first step, we decided to extend the JJGW07 in vitro profile evaluation and verify its binding to 5-HT_6_ receptors. Many studies have indicated that these receptors are involved in the pathophysiology of anxiety and depression, and might be potential targets to treat these disorders [9,29,30]. Moreover, there is much evidence that neurotransmission via the 5-HT_6_ receptors in the brain is involved in learning processes, and the blockade of these receptors results in a significant improvement to cognitive function [31,32,33,34,35,36]. Many antipsychotics, such as clozapine or olanzapine, antagonize 5-HT_6_ receptors in the brain and possess procognitive properties, probably due to this mechanism [37]. However, radioligand binding studies have shown no affinity of JJGW07 for 5-HT_6_ receptors, suggesting that other biological targets are involved in its pharmacological effects. Importantly, the memory-enhancing activity is not only dependent on the modulation of 5-HT_6_ receptors, as there are other antipsychotics, i.e., aripiprazole, brexpiprazole, or cariprazine, that show procognitive properties due to other mechanisms, including interactions with the 5-HT_1A_ and/or 5-HT_7_ receptors [38,39,40].

Next, we performed functional studies to determine the type of interaction with the 5-HT_1A_, 5-HT_2A_, 5-HT_7_, and D_2_ receptors. JJGW07 showed strong antagonistic properties at dopamine D_2_ and serotonin 5-HT_1A_ receptors, whereas it showed weak antagonistic effects towards serotonin 5-HT_2A_ and 5-HT_7_ receptors. The obtained results agree with our previous studies of JJGW07 analog, which also showed antagonistic properties for serotonin 5-HT_1A_, 5-HT_2A_, and 5-HT_7_ and dopamine D_2_ receptors [6].

Bearing in mind the significant role of the serotonin system, especially 5-HT_1A_ and 5-HT_7_ receptor modulation, in depression [9], we decided to evaluate the antidepressant-like effect of the tested compound in the forced swim and tail suspension tests in mice, which are two common behavioral tests to assess the antidepressant-like activity of novel compounds. JJGW07 was active only in the tail suspension test, which has very good predictive validity [41]. The observed results may be due to the differences between procedures in both tests, i.e., the environment and the risk of hypothermia in the forced swim test or the duration of the experiments. JJGW07 decreased the immobility of animals at a dose around 133-fold lower than that of moclobemide [42]. Notably, JJGW07 did not affect the spontaneous locomotor activity of mice at the antidepressant-like doses, which indicated that these results are not associated with the psychostimulant properties of the compound.

In the next step of our study, we investigated the anxiolytic-like activity of the tested compound in the four-plate, elevated plus maze, and marble-burying tests in mice. The modulation of the serotonin system, especially the blockade of 5-HT_1A_ and 5-HT_7_ receptors, is involved in the anxiolytic-like effects of many compounds [22,43,44,45,46,47,48,49,50,51]. The four-plate test is one of the most common tests for assessing the anxiolytic-like properties of novel compounds, and is based on the conflict situation when an electric shock inhibits a mouse’s natural tendency to explore a new environment. Our study showed that JJGW07 possessed the anxiolytic-like activity, and the effect was comparable to diazepam [44]. However, the anxiolytic-like activity of JJGW07 was observed only in the medium-tested dose (inverted U-shaped dose effect). This non-linear response is a common phenomenon of compounds with anxiolytic-like properties, and it needs further examination to explain it because of its multifactor impact. To confirm the obtained results, we performed two additional experiments—the marble-burying test and the elevated plus maze test. As described in the literature, anxiolytics decrease the number of buried marbles by rodents, which is identified as a reduction in natural “defensive burying” in response to an aversive stimuli for novelty [52]. We demonstrated the possible anxiolytic-like properties of JJGW07 in this test in mice, and similarly, we observed an inverted U-shaped dose effect (middle doses were effective). However, the marble-burying test may indicate not only anxiolytic activity, but also antidepressant properties of the tested compound [52]; as well, it can reflect obsessive–compulsive-like behavior [52,53]. Moreover, disturbances in the hippocampus function may also affect the typical behavior of rodents, observed as a reduction in the burrowed activity in hippocampal-lesioned mice [53]. Furthermore, Thomas and colleagues demonstrated that the marble-burying behavior is neither correlated with anxiety-like traits nor stimulated by novelty, but is rather genetically regulated [54]. Bearing that in mind, we should be cautious when interpreting the marble-burying test results. The third test assessing the anxiolytic-like properties was the elevated plus maze test, based on the rodent’s natural tendency to avoid open spaces. The tested compound did not increase the time spent in open arms, nor the number of entries into open arms. Conversely, JJGW07 significantly reduced the number of open arm entries. However, since the compound showed anxiolytic-like properties in the four-plate and marble-burying tests, this effect was likely not due to the anxiogenic potential. Nonetheless, this issue requires further studies. Significantly, JJGW07 at all anxiolytic-like doses did not affect the locomotor activity of mice; thus, we can assume that the observed effects are specific.

Compounds that act on the central nervous system, such as anxiolytics or antidepressants, may affect the motor function negatively [55,56,57,58]. Therefore, we investigated the influence of JJGW07 on the motor coordination in mice. The tested compound did not impair motor coordination at the antidepressant-like and anxiolytic-like doses. Furthermore, a negative effect on the motor coordination was observed at a dose about 189-fold higher than the lowest active dose in the behavioral tests. These results suggest that JJGW07 did not induce motor disturbances at the pharmacologically active doses.

Since the tested compound possessed antagonistic properties not only at serotonin receptors, but also at dopamine D_2_ receptors, we evaluated its potential antipsychotic activity. The blockage of dopamine D_2_ receptors is one of the primary mechanisms of action among commonly used antipsychotic drugs, especially first-generation neuroleptics [59]. Simultaneously, atypical neuroleptics (i.e., aripiprazole and cariprazine) act via both dopamine and serotonin receptors, mostly D_2_, D_3_, 5-HT_2A_, and 5-HT_1A_ receptors [60,61,62,63]. Therefore, to assess the antipsychotic-like activity of JJGW07, we used the MK-801-induced hyperlocomotion test. MK-801, as a non-competitive NMDA receptor antagonist, increases dopamine transmission in different brain areas (i.e., medial prefrontal cortex and nucleus accumbens) [64], which, in animal behavior, manifests mainly as hyperlocomotion and circling. Therefore, compounds that decrease this hyperactivity are likely to have antipsychotic-like properties. JJGW07 reversed the MK-801-induced hyperlocomotion in mice at doses of 0.625–2.5 mg/kg. These results encouraged us to investigate the effects of JJGW07 in another test, using amphetamine as an inductor of excessive mobility. Amphetamine stimulates not only dopamine, but also noradrenaline neurotransmission, resulting in the hyperlocomotion of animals [65]. However, JJGW07 did not reduce the amphetamine-induced hyperlocomotion in mice. Significantly, the tested compound administered alone did not influence the locomotor activity at the active doses, suggesting no psychostimulant properties. Based on these results, we may assume that the antipsychotic-like effects of JJGW07 were due to dopamine rather than noradrenaline neurotransmission modulation, but finding out the exact mechanisms of the JJGW07 antipsychotic-like effect requires further experiments.

Unfortunately, treatment with antipsychotic drugs is often associated with extrapyramidal side effects due to the blockage of dopamine D_2_ receptors [66,67,68,69]. For that reason, as the next step of our study, we assessed the risk of JJGW07 to induce catalepsy in mice. A cataleptic state is a specific symptom of schizophrenia, which manifests in rodents as a numb body and the maintenance of an abnormal body position for a more extended period. Our results indicated that JJGW07 did not induce catalepsy at the antipsychotic-like doses. Moreover, the cataleptogenic dose of the tested compound was 16-fold higher than the lowest antipsychotic-like dose. Thus, we suspect that JJGW07 possesses a low risk of inducing extrapyramidal side effects. However, further studies are required to assess its safety, especially after chronic administration.

Patients with psychiatric and mood disorders very often suffer from cognitive deficits. Learning and memory impairments occur in patients with schizophrenia [70,71,72] as well as in depressed patients and people with anxiety [73,74,75,76,77]. Therefore, we examined the potential procognitive and anti-amnesic effects of JJGW07 in the passive avoidance task in mice. This test is used to assess the influence of novel compounds on the emotional memory in rodents, as it is based on conflicting behavior, i.e., the avoidance of electric stimuli vs. the natural tendency to avoid open spaces and enter dark chambers with aversive stimuli. JJGW07 given alone at the doses of 0.625–2.5 mg/kg increased the latency time to enter the dark compartment, which suggests that it did not impair long-term memory in mice. These results are comparable to the effects of aripiprazole, which showed memory-enhancing properties.

Furthermore, using the same test, we evaluated the ability of JJGW07 to protect against MK-801-induced memory disruptions in mice. The administration of MK-801, an NMDA receptor antagonist, resulted in memory impairments, including long-term memory deficits. Our results indicated that JJGW07 ameliorated the memory deficits induced by MK-801, and this effect was not observed in the case of aripiprazole. Interestingly, JJGW07 displayed an inverted U-shaped dose–effect curve, as only the medium-tested dose was active. Although this non-linear relationship is prevalent in neuropharmacology, it is still difficult to explain. In the case of JJGW07, it might be due to its interaction with several receptors, which can be differentially influenced at various doses.

Encouraged by the positive influence on emotional memory, as the final step of our study, we investigated the effect of JJGW07 on recognition memory using the object recognition test in mice. This test is based on the natural tendency of animals to explore a novel object rather than those previously learned [78]. JJGW07 at all tested doses impaired the recognition memory in mice, whereas aripiprazole did not affect recognition memory acquisition at all tested doses. Moreover, aripiprazole protected mice from the memory deficits induced by MK-801 only at the lowest tested doses, i.e., 0.25 and 0.5 mg/kg. Our studies suggest that JJGW07 might improve long-term emotional, but not recognition memory, deficits. Further experiments are required to better assess the tested compound’s influence on learning and memory processes.

Interestingly, the pharmacological profile of JJGW07 is very similar to its chemical derivative, JJGW08—a compound with the antipsychotic-like and the anxiolytic-like activity in rodents [6], as well as the anti-amnesic properties in mice (manuscript under review). Both compounds are antagonists of serotonin 5-HT_1A_, 5-HT_2A_, and 5-HT_7_ and dopamine D_2_ receptors, with no affinity towards 5-HT_6_ receptors. The advantage of JJGW07 is an additional antidepressant-like activity demonstrated in the tail suspension test in mice. In contrast, JJGW08 showed the antipsychotic-like and the anxiolytic-like effects over a broader range of doses than JJGW07 (0.15–2.5 mg/g vs. 0.625–2.5 mg/kg and 0.3–1.25 mg/kg vs. 0.625–1.25 mg/kg, respectively). Nevertheless, it is worth further investigating the tested compound.

There are some limitations of our study. First, we assessed the activity of the tested compound only after a single administration. The chronic administration of JJGW07, using animal models of depression, anxiety, and schizophrenia, would give insight into the tested compound’s safety profile and long-term pharmacological effects. Moreover, as not only D_2_ receptors, but also other types of dopamine receptors are involved in the pathogenesis of schizophrenia, we should also evaluate the affinity of the tested compound for a full panel of dopamine receptors. Furthermore, we should extend our study on the receptor profile of JJGW07 involving its activity towards NMDA receptors, as we used MK-801, the NMDA receptor antagonist, in several behavioral tests. Therefore, our future research will be focused on the extensive pharmacological characterization of JJGW07.

## 4. Materials and Methods

### 4.1. Drugs

The studied compound 2-{5-[4-(2-methoxyphenyl)piperazin-1-ylo]butoxy}benzamide hydrochloride, JJGW07, was synthesized in the Department of Organic Chemistry and Technology, Faculty of Chemical and Engineering and Technology, Cracow University of Technology. The synthesis and biological properties of the compound were described earlier [5].

JJGW07 was dissolved in saline (0.9 % NaCl, Polpharma, Starogard Gdańsk, Poland) and administered intraperitoneally (*ip*) 30 min before each behavioral test. The chemicals used in the radioligand studies, i.e., methiothepin (Sigma-Aldrich, Darmstadt, Germany), were dissolved in saline. MK-801 (Sigma-Aldrich, Darmstadt, Germany) was dissolved in saline and administered *ip* 15 min before experiments, whereas amphetamine (Sigma-Aldrich, Darmstadt, Germany) was dissolved in saline and administered subcutaneously (*sc*) 30 min before tests. Aripiprazole (Sigma-Aldrich, Darmstadt, Germany) was dissolved in 1% Tween (J.T. Baker, Phillipsburg, NJ, USA) and administered *ip* 30 min before experiments. The control groups received 0.9% NaCl solution or 1% Tween as a vehicle. The tested compound was administered at the dose range of 0.625–2.5 mg/kg. The dose range was chosen based on our previous experiments on salicylamide derivatives [6]. If the effect of the lowest dose, i.e., 0.625 mg/kg, was still statistically significant, we decreased the dose by half until antidepressant-like, anxiolytic-like, antipsychotic-like, or anti-amnesic activity disappeared.

The two exceptions were the effect on motor coordination and the cataleptogenic potential of JJGW07, as to perform those experiments, we needed to use higher doses, i.e., 5–40 mg/kg.

### 4.2. Animals

All experiments were performed on adult male CD-1 mice, weighing 18–21 g, obtained from an accredited house at the Faculty of Pharmacy, Jagiellonian University Medical College, Krakow, Poland. Animals were kept in groups of 10 mice in plastic standard cages (37 cm × 21 cm × 15 cm) in a controlled environment (i.e., constant room temperature (22 ± 2 °C), adequate humidity (40–60%), and a 12 h light/dark cycle), with ad libitum access to food and water. Behavioral procedures were performed between 8 a.m. and 4 p.m. by a trained observer blind to the treatments. Animals were selected randomly for the treatment groups. Each group consisted of 8–10 mice that were used only once in each test and were euthanized immediately after the experiments. All injections were administered in a 10 mL/kg volume. Procedures involving animals were conducted according to the current European Community and Polish legislation on animal experimentation.

### 4.3. Radioligand Binding Assay

Radioligand binding assays were performed using membranes from CHO-K1 cells, which were stably transfected with the human 5-HT_6_ receptor. The assay procedures were conducted according to a slightly modified method described by Sałaciak and colleagues [79].

Binding experiments were conducted in 96-well microplates, and the reaction mix included a solution of the test compound, radioligand, and diluted membranes or the tissue suspension. Specific assay conditions for each target are shown in Table 5. The reaction was terminated by rapid filtration through a GF/B or GF/C filter mate using an automated harvester system, FilterMate Harvester (PerkinElmer, Boston, MA, USA). The filter mates were dried at 37 °C in a forced-air fan incubator, and then the solid scintillator MeltiLex was melted on the filter mates at 90 °C for 5 min. Radioactivity was counted in the MicroBeta2 scintillation counter (PerkinElmer, Boston, MA, USA) at approximately 30% efficiency. The concentrations of the analyzed compounds ranged from 10^–10^ to 10^–5^ M. The inhibitory constant (Ki) was estimated using GraphPad Prism 5.0 (GraphPad Software, San Diego, CA, USA). A single assay was performed with each compound concentration in duplicate, and the whole assay was repeated in three independent experiments. Inhibition constants (Ki) were calculated according to the equation of Cheng and Prusoff [80].

### 4.4. Functional Assay for 5-HT_1A_, 5-HT_2A_, and D_2_ Receptor

Tested and reference compounds were dissolved in dimethyl sulfoxide (DMSO) at a concentration of 10 mM. Serial dilutions were prepared in a 96-well microplate in an assay buffer, and 8 to 10 concentrations were tested. An intrinsic activity assay was performed according to the manufacturer of the ready-to-use CHO-K1 cells with the stable expression of the human serotonin 5-HT_1A_, 5-HT_2A_, and D_2_ receptors, human GPCR, the promiscuous G protein Gαqi/5 for the D_2_ receptor, and α_16_ for 5-HT_1A_ and 5-HT_2A_ (Perkin Elmer, Boston, MA, USA). The assay was executed according to a previously described protocol [81]. After thawing, cells were transferred to the assay buffer (DMEM/HAM’s F12 with 0.1% protease-free BSA) and centrifuged. The cell pellet was resuspended in the assay buffer, and coelenterazine h was added at a final concentration of 5 µM. The cell suspension was incubated at 16 °C (or 21 °C), protected from light with constant agitation for 16 h (or 4 h), and then diluted with the assay buffer to the concentration of 100,000 cells/mL (or 250,000 cells/mL). After 1 h of incubation, 50 µL of the cell’s suspension was dispensed using automatic injectors built into the radiometric and luminescence plate counter MicroBeta2 LumiJET (PerkinElmer, Boston, MA, USA) into white opaque 96-well microplates preloaded with test compounds. The immediate light emission generated following calcium mobilization was recorded for 30 s. In antagonist mode, after 25–30 min of incubation, the reference agonist was added to the above assay mix, and the light emission was recorded again. The final concentration of the reference agonist (100 nM serotonin for the 5-HT_1A_ receptor, 30 nM α-methylserotonin for the 5-HT_2A_ receptor, and 30 nM apomorphine for the D_2_ receptor) was equal to EC_80_. IC_50_ and EC_50_ values were calculated.

### 4.5. Functional Assays for 5-HT_7_ Receptors

Test and reference compounds were dissolved in DMSO at a concentration of 1 mM. Serial dilutions were prepared in a 96-well microplate in the assay buffer and 8 to 10 concentrations were tested. For the 5-HT_7_ receptor, the adenylyl cyclase activity was monitored using cryopreserved CHO-K1 cells with expression of the human serotonin 5-HT_7_ receptor. A functional assay based on cells with expression of the human 5-HT_7_ receptor was performed, according to a previously described protocol [44]. CHO-K1 cells were transfected with a beta-lactamase (bla) reporter gene under the control of the cyclic AMP response element (CRE) (Life Technologies, Carlsbad, CA, USA).

Thawed cells were resuspended in a stimulation buffer (HBSS, 5 mM HEPES, 0.5 IBMX, and 0.1% BSA at pH 7.4) at 200,000 cells/mL. The same volume (10 μL) of cell suspension was added to tested compounds loaded onto a white opaque half-area 96-well microplate. The antagonist response experiment was performed with 10 nM serotonin as the reference agonist. The agonist and antagonist were added simultaneously. Cell stimulation was performed for 60 min at room temperature. After incubation, cAMP measurements were performed with a homogeneous TR-FRET immunoassay using the LANCE Ultra cAMP kit (PerkinElmer, Boston, MA, USA). Then, 10 μL of EucAMP Tracer Working Solution and 10 μL of ULight-anti-cAMP Tracer Working Solution were added, mixed, and incubated for 1 h. The TR-FRET signal was read on an EnVision microplate reader (PerkinElmer, Boston, MA, USA). IC_50_ and EC_50_ values were calculated by a non-linear regression analysis using GraphPad Prism 5.0 software. The log IC_50_ was used to obtain the K_b_ by applying the Cheng–Prusoff approximation.

### 4.6. Forced Swim Test

The experiment was performed on mice according to the method described by Porsolt et al., and as previously described [82,83]. Mice were placed individually in glass cylinders (height: 25 cm, diameter: 10 cm) filled with water at 24 ± 1 °C to a depth of 10 cm and were left there for 6 min. Following a 2 min habituation period, the total time spent immobile was recorded during the next 4 min. The animal was regarded as immobile when it remained floating passively in the water, making only small movements to keep its head above the water. The experiments were video-recorded and scored using elevenmaze.com software by a trained observer blind to the treatment.

### 4.7. Tail Suspension Test

The experiment was carried out on mice according to the method described by Steru et al., and as previously described [42,84]. The mice were suspended by their tails using a medical adhesive tape at a height of 50 cm above a flat surface, in such a position that they could not escape or hold on to nearby surfaces. The total time of immobility was measured during the 6 min test period. Immobility was defined as the animal hanging passively without limb movement. The experiments were performed by a trained observer blind to the treatment.

### 4.8. Four-Plate Test

The four plate test was performed on mice according to a previously described method [85,86]. Mice were placed individually in a four-plate apparatus. After a 15 s habituation period, each mouse that crossed from one plate to another (two limbs on one plate, two on another) was punished by an electric shock (0.8 mA, 0.5 s). The number of punished crossings was calculated during the 60 s of the test.

### 4.9. Marble-Burying Test

The test was conducted according to the method described by Broekkamp et al. [87] with minor modifications. Mice were placed individually in plastic cages, identical to their home cages, that contained a layer of bedding and 20 glass balls (1.6 cm in diameter) arranged in a pattern of 4 × 5. After 30 min of the experiment, the mice were removed from the cages, and the number of balls buried to at least 2/3 of their size was counted. A reduction in the number of buried balls compared to the control group suggested the tested compound’s anxiolytic-like properties.

### 4.10. Elevated Plus Maze Test

The elevated plus maze was performed according to a method previously described [18,19]. The apparatus consisted of two opposing open (30 cm × 5 cm) and two enclosed (30 cm × 5 cm × 25 cm) arms connected by a central platform, forming the shape of a plus sign. The open and closed arms were connected by a central field (5 cm × 5 cm). Each mouse was individually placed in the central field of the apparatus with its head turned toward one of the closed arms. Animal behavior was observed for 5 min. The device was disinfected with an odorless disinfection solution after each mouse. The number of entries to open and closed arms and the time spent in the open arms were measured. The experiments were recorded and scored using aLab.io software by a trained observer blind to the treatments.

### 4.11. Spontaneous Locomotor Activity in Mice

The locomotor activity of mice was measured as previously described [44]. The mobility of the animals was measured in actometers, i.e., in plastic Opto M3 cages (22 × 12 × 13 cm) connected to a computer with MultiDevice Software v.1.30 (Columbus Instruments, Columbus, OH, USA). The experimental cages were equipped with infrared sources on one side and sensors receiving the emitted rays on the other side of the cage. The crossing of each beam of infrared rays was classified as the motor activity. Each mouse was placed individually in a cage for the 30 min habituation period (directly after administration of the studied compound), and then the number of photobeam crossings was recorded (ambulation). Locomotor activity was evaluated every 1 or 5 min for 1–60 min depending on the observation period in the behavioral tests (60 min for the hyperlocomotion test, 30 min for the marble-burying test, 6 min for the tail suspension test, 5 min for the elevated plus maze test, 4 min for the forced swim test, and 1 min for the four-plate test). The cages were disinfected with the odorless disinfection solution after each mouse.

### 4.12. Rotarod Test

The experimental procedure has been described in detail by Pytka K. et al. [88]. Mice were trained on a rotarod apparatus (May Commat RR0711, Ankara, Turkey; rod diameter: 2 cm) for 3 consecutive days. During each training session, animals were placed for 3 min on the rotating rod (24 rpm, constant speed) with an unlimited number of trials. The experiment was performed 24 h after the last training session. On the test day, mice were injected with the studied compounds, and 30 min later, placed on the rotarod. The criterion of motor impairments was the inability of the animal to remain on the rotating rod for 60 s. The TD_50_ value was calculated as the dose at which 50% of the animals could not stay on the rotating rod [27].

### 4.13. MK-801- and Amphetamine-Induced Hyperlocomotion Test in Mice

The test was performed according to the method described by Carlsson et al. [89,90], using the same apparatus as described in Section 4.11. The animals were placed individually in experimental cages immediately after the administration of the tested compound, 30 min before the start of the test, to adapt to the new conditions and exclude the occurrence of hyperactivity caused by a change in the environment. Spontaneous locomotor activity was measured every 5 min for 60 min. Mice received two injections: one of the tested compound (30 min before the test, *ip*) and one of amphetamine (2.5 mg/kg, 30 min before the test, *sc*) or MK-801 (0.2 mg/kg, 15 min before the test, *ip*). Control groups received either an injection of saline and amphetamine or MK-801, or two injections of saline.

### 4.14. Catalepsy Bar Test

Catalepsy was assessed using the bar method described by Ueki et al. with minor modifications [91,92]. The front paws of the mice were placed on a cylindrical metal bar located 4 cm above the tabletop’s surface, while the hind paws remained on the tabletop. The time for which the animal held both paws on the bar was measured at 30, 60, and 120 min after the administration of the tested compound, with a maximum measurement time of 60 s. Each of the measurements consisted of placing the animal’s paws on the bar three times unless the mouse was on it for 60 s; then, no further trial was performed. The score for each trial was assessed as follows [28]:-Score of 0 points if the animal held the constrained position for <15 s;-Score of 1 point if the animal stayed on the bar for 15–29.9 s;-Score of 2 points if the animal stayed on the bar for 30–59.9 s;-Score of 3 points if the animal stayed on the bar for more than 60 s.

### 4.15. Step-Through Passive Avoidance Task

The step-through passive avoidance task was performed according to a method previously described [45,93]. The apparatus for the step-through passive avoidance task consisted of two compartments (i.e., bright and dark) separated by an automated sliding door (LE872, Bioseb, Vitrolles, France). For the acquisition session, mice were placed individually in a bright compartment (20 cm × 21 cm × 20 cm, 1000 lx) with a closed door to a smaller, dark compartment (7.3 cm × 7.5 cm × 14 cm, 10 lx) equipped with an electric grid floor. A total of 30 s after placing the animal in the bright compartment, the door to the dark compartment was opened. If the mouse entered the dark compartment, the door closed immediately, and the rodent was punished by an electric foot shock (0.8 mA for 2 s). The mice that did not enter the dark compartment within the next 50 s were excluded from the study. On the following day (24 h later), animals were placed again in the bright compartment for 300 s (retention session) with the difference that after entering the dark compartment, the mice did not receive an electric shock. The latency to enter the dark compartment was measured. The tested and reference compounds were administered *ip* 30 min before the acquisition trial. To induce memory impairments, MK-801 (0.125 mg/kg) was administered *ip* 15 min before the experiment. Control groups were injected *ip* with saline/1% Tween or with saline/1% Tween and MK-801.

### 4.16. Object Recognition Test

The test was performed according to the method described earlier [78,94] and consisted of a familiarization and habituation session. In the familiarization session, mice were placed individually in a box (35 cm × 35 cm × 35 cm) and remained there until the total exploration time for two identical objects (2 towers of Lego bricks or 2 bottles filled with sand) was 20 s or until 10 min has passed. Animals that did not meet the criteria were eliminated from further study. After 24 h, the mice were placed again in the box, with the difference that one object was replaced with a new one. As before, mice remained in the open field until the total exploration time reached 20 s, but not for longer than 10 min. In this session, the exploration time for each object was measured separately. The tested and reference compounds were administered *ip* 30 min before the familiarization phase to assess their influence on recognition memory. To induce memory disturbances, MK-801 (0.125 mg/kg) was administered *ip* 15 min before the experiment. Control groups were injected *ip* with saline/1% Tween or with saline/1% Tween and MK-801.

### 4.17. Data Analysis

The results are presented as means ± SD (standard deviation) or medians with an interquartile range (nonparametric analysis). The normality of data sets and their homogeneity were determined using D’Agostino and Pearson and Brown–Forsythe tests, respectively. The comparisons between experimental and control groups were performed by one-way ANOVA followed by a Newman–Keuls post hoc test or two-way ANOVA with repeated measures followed by a Bonferroni post hoc test. A one-sample *t*-test was used to analyze the results of the object recognition task. In cases when assumptions for a normal distribution of data were not fulfilled, we used a Kruskal–Wallis with Dunn’s post hoc test. A value of *p* < 0.05 was considered to be significant.

## 5. Conclusions

In this study, we evaluated the basic pharmacological profile of JJGW07, a novel arylpiperazine alkyl derivative of salicylamide, which demonstrated strong antagonistic properties at dopamine D_2_ and serotonin 5-HT_1A_ receptors, very weak properties at serotonin 5-HT_2A_ and 5-HT_7_ receptors, and no affinity towards 5-HT_6_ receptors. The tested compound showed antidepressant-, anxiolytic-, and antipsychotic-like properties in rodents. Furthermore, JJGW07 did not affect the locomotor coordination or induce catalepsy in mice at antipsychotic-like doses. Moreover, JJGW07 displayed procognitive and anti-amnesic properties in mice. Overall, our observations suggest that JJGW07 could be a model structure for the synthesis of new derivatives of salicylamide with potential use in the treatment of psychiatric disorders such as depression, anxiety, schizophrenia, and memory deficits.

## Figures and Tables

**Figure 1 pharmaceuticals-16-00175-f001:**
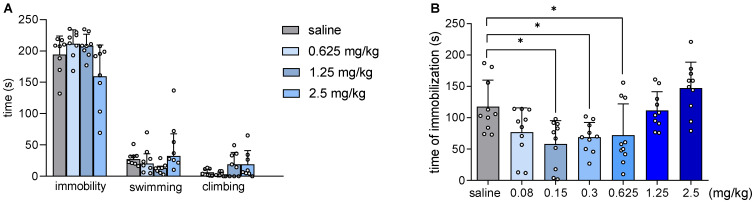
The influence of JJGW07 on the immobility, swimming, and climbing times in the forced swim test (panel (**A**)) and the immobility time in the tail suspension test (panel (**B**)) in mice. In the forced swim test, mice were placed in water tanks and the immobility time, the swimming time, and the climbing time were measured for 4 min (after a 2 min adaptation period), whereas in the tail suspension test, mice were suspended by the tail 50 cm above a flat surface with an adhesive tape and the immobility time was measured for 6 min. JJGW07 was administered intraperitoneally (*ip*) 30 min before the test. The control group received an injection of 0.9% NaCl (*ip*). Values are expressed as means ± SD or medians with interquartile range; n = 8–10 mice per group. Statistical analysis: one-way ANOVA (Newman–Keuls post hoc) and Kruskal–Wallis test (Dunn post hoc); * *p* < 0.05 vs. control group.

**Figure 2 pharmaceuticals-16-00175-f002:**
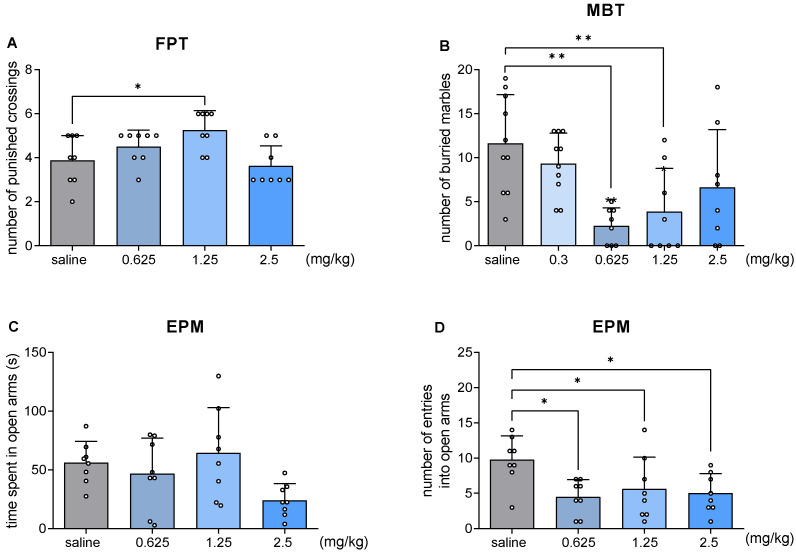
The influence of JJGW07 on the number of punished crossings (Panel (**A**)), the number of buried marbles (Panel (**B**)), and time spent in the open arms (Panel (**C**)), as well as the number of entries into open arms (Panel (**D**)). (**A**) Mice were placed in the four-plate apparatus, and after a 15 s adaptation period, each crossing from one plate to another was punished by an electric shock. The number of punished crossings was measured for 60 s. (**B**) Mice were placed individually in cages with a 5 cm bedding layer, where 20 glass balls were placed. The number of buried marbles after 30 min of the test was counted. (**C**,**D**) In the elevated plus maze test, mice were placed individually in the center area of the maze and observed for 5 min. The number of entries into the open arms, as well as the time spent in the open arms, was measured. JJGW07 was administered intraperitoneally (*ip*) 30 min before the test. The control group received an injection of 0.9% NaCl (*ip*). Values are expressed as means ± SD; n = 8 mice per group. Statistical analysis: one-way ANOVA (Newman–Keuls post hoc), * *p* < 0.05, ** *p* < 0.01 vs. control group, FPT—the four-plate test, MBT—the marble-burying test, and EPM—the elevated plus maze test.

**Figure 3 pharmaceuticals-16-00175-f003:**
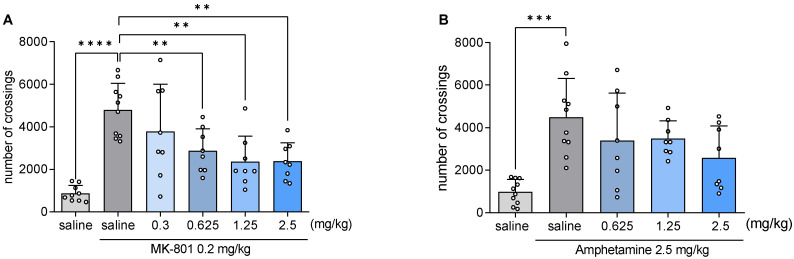
The effect of JJGW07 on the MK-801-induced (Panel (**A**)) and amphetamine-induced (Panel (**B**)) hyperlocomotion in mice. Locomotor activity was recorded in actometers separately for each mouse. After 30 min of adaptation, the number of crossings of photobeams was measured during 60 min. JJGW07 was administered intraperitoneally (*ip*) 30 min before the test. MK-801 (0.2 mg/kg, *ip*) was administered 15 min before the experiment, while amphetamine (2.5 mg/kg) was administered subcutaneously (*sc*) 30 min before the experiment. The control groups received either two injections of 0.9% NaCl (*ip* or *sc*), or 0.9% NaCl *(ip)* and MK-801 (*ip*), or 0.9% NaCl *(ip)* and amphetamine (*sc*). Values are expressed as means ± SD; n = 8–10 mice per group. Statistical analysis: one-way ANOVA (Newman–Keuls post hoc); ** *p* < 0.01, *** *p* < 0.001, **** *p* < 0.0001.

**Figure 4 pharmaceuticals-16-00175-f004:**
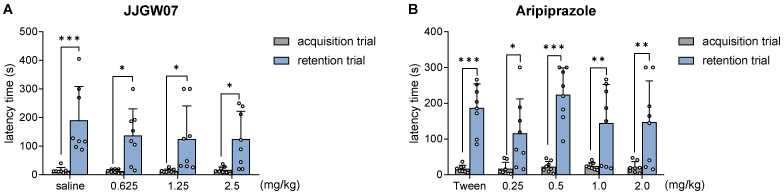
The influence of JJGW07 (Panel (**A**)) and aripiprazole (panel (**B**)) on the latency in the step-through passive avoidance task in mice. The experiment consisted of two sessions. In the acquisition trial, mice were placed individually in the light chamber of the apparatus, with the door opening after 30 s. When the animal crossed to the dark chamber, the door closed and the animal was punished with an electric shock (0.8 mA, 2 s). JJGW07 and aripiprazole were administered intraperitoneally (*ip*) 30 min before the start of the experiment. The control group received an *ip* 0.9% NaCl solution or 1% Tween. On the second day of the test, the mice were placed again in a bright chamber and the latency was measured for a maximum of 300 s (without the electrical impulse). Values are expressed as means ± SD; n = 8 mice per group. Statistical analysis: two-way ANOVA with repeated measures (Bonferroni post hoc); * *p* < 0.05, ** *p* < 0.01, *** *p* < 0.001 vs. acquisition trial.

**Figure 5 pharmaceuticals-16-00175-f005:**
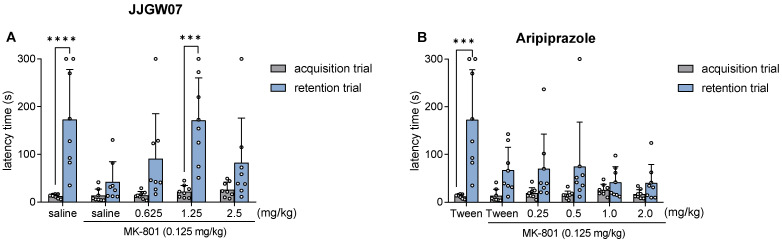
The influence of JJGW07 (Panel (**A**)) and aripiprazole (Panel (**B**)) on the latency after the MK-801 administration in the step-through passive avoidance task in mice. The experiment consisted of two sessions. In the acquisition trial, mice were placed individually in the light chamber of the apparatus, with the door opening after 30 s. When the animal crossed to the dark chamber, the door closed and the animal was punished with an electric shock (0.8 mA, 2 s). JJGW07 and aripiprazole were administered intraperitoneally (*ip*) 30 min before the test, while MK-801 (0.125 mg/kg) was administered *ip* 15 min before the start of the experiment to induce the memory impairments. The control group received an *ip* 0.9% NaCl/1% Tween solution in two injections or 0.9% NaCl/1% Tween solution and MK-801 (0.125 mg/kg; *ip*). On the second day of the test, the mice were placed again in a bright chamber and the latency was measured for a maximum of 300 s (without the electrical impulse). Values are expressed as means ± SD; n = 8 mice per group. Statistical analysis: two-way ANOVA with repeated measures (Bonferroni post hoc); *** *p* < 0.001, **** *p* < 0.0001 vs. acquisition trial.

**Figure 6 pharmaceuticals-16-00175-f006:**
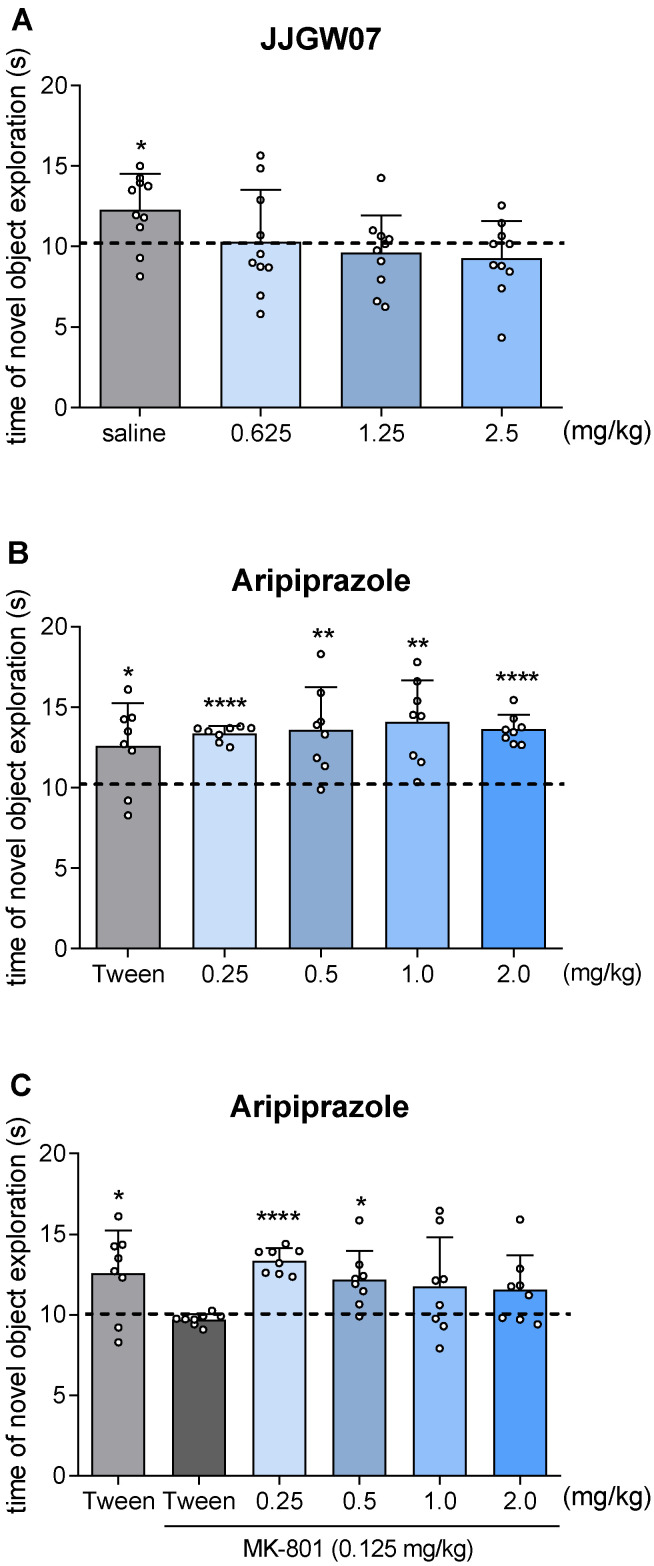
The influence of JJGW07 (Panel (**A**)) and aripiprazole (Panel (**B**)) on the novel object exploration time in mice as well as the influence of aripiprazole (Panel (**C**)) on the novel object exploration time after the MK-801 administration in mice. The experiment consisted of two sessions. On the first day, mice were placed and left in cages until they reached a total exploration time of 20 s for both identical subjects, but for no longer than 10 min. JJGW07 and aripiprazole were administered intraperitoneally (*ip*) 30 min before the start of the experiment, while MK-801 (0.125 mg/kg) was administered *ip* 15 min before the start of the experiment to induce the memory impairments. The control group received an *ip* 0.9% NaCl solution (**A**) or 1% Tween (**B**) as well as two injections of 1% Tween or 1% Tween (*ip*) and MK-801 (0.125 mg/kg; *ip*) (**C**). On the second day, mice were placed again in cages, where one subject was changed to a new one. The mice remained in the cage until they reached a total exploration time of 20 s for both objects, but for no longer than 10 min. Values are expressed as means ± SD; n = 8–10 mice per group. Statistical analysis: one-sample *t*-test; * *p* < 0.05, ** *p* < 0.01, **** *p* < 0.0001 vs. chance level = 10 s.

**Table 1 pharmaceuticals-16-00175-t001:** The intrinsic activity of JJGW07 for dopamine D_2_ and serotonin 5-HT_1A_, 5-HT_2A_, and 5-HT_7_ receptors.

Receptor	Treatment	Agonist Mode		Antagonist Mode			
		E_max_%	*p*EC_50_ ± Range	E_max_%	*p*IC_50_ ± Range	K_b_ (nM)	R^2^K_b_
	Quiniprol	100	8.70 ± 0.12	0	n.c.	n.c.	n.c.
D_2_	Apomorphine	100	7.50 ± 0.08	0	n.c.	n.c.	n.c.
	Chlorpromazine	2	n.c.	0	9.78 ± 0.42	0.03	0.94
	JJGW07	20	n.c.	3	7.03 ± 0.01	18	0.86
	Serotonin	100	7.63 ± 1.05	0	n.c.	n.c	n.c.
5-HT_1A_	NAN-190	6	n.c.	0	8.98 ± 0.05	0.07	0.99
	JJGW07	1	n.c	0	7.38 ± 0.52	3	0.85
	α-methyloserotonin	100	8.50 ± 0.31	2	n.c.	n.c.	n.c.
5-HT_2A_	Serotonin	112	8.36 ± 0.04	1	n.c.	n.c.	n.c.
	Mianserin	3	n.c.	3	8.07 ± 0.08	2.3	0.91
	JJGW07	28	5.81 ± 0.29	17	5.56 ± 0.87	550	0.92
	Serotonin	100	8.06 ± 0.14	0	n.c.	n.c.	n.c.
5-HT_7_	SB-269970	0	n.c.	9.0	9.29 ± 0.21	0.2	0.94
	JJGW07	1	n.c.	6.0	5.46 ± 0.34	1600	0.99

Data are expressed as the means ± range of two independent experiments in duplicate. E_max_—the maximum possible effect; *p*EC_50_—the negative logarithm of the concentration of a compound where 50% of its maximal effect was observed; *p*IC_50_—the logarithm of the concentration of a compound where 50% of its maximal inhibitory effect was observed; K_b_—the equilibrium dissociation constant of a competitive antagonist determined using the Cheng equation; and R^2^—the coefficient of determination [26]. n.c.—non-calculable.

**Table 2 pharmaceuticals-16-00175-t002:** The influence of JJGW07 on the locomotor activity in mice.

Treatment	Dose (mg/kg)	Number of Crossings ± SD																	
		60 min			30 min			6 min			5 min			4 min			1 min		
Saline	-	2112	±	1156	773	±	578	274	±	106	198	±	102	184	±	81	38	±	24
	0.08		-			-		226		37		-			-			-	
	0.15		-			-		325	±	84		-			-			-	
JJGW07	0.3		-		1133	±	224	305	±	67		-			-			-	
	0.625	2247	±	362	806	±	289	237	±	49	193	±	35	161	±	51	32	±	11
	1.25	2029	±	781	1248	±	441	232	±	75	203	±	73	141	±	78	35	±	20
	2.5	3035	±	962	884	±	262	270	±	50	214	±	62	179	±	72	42	±	20

Locomotor activity was recorded separately for each mouse in actometers. After the 30 min adaptation period, the number of photobeam crossings was measured at the appropriate time intervals, i.e., 60 min for the hyperlocomotion test, 30 min for the marble-burying test, 6 min for the tail suspension test, 5 min for the elevated plus maze test, 4 min for the forced swim test, and 1 min for the four-plate test. JJGW07 was administered intraperitoneally (*ip*) 30 min before the test. The control group received an injection of 0.9% NaCl (*ip*). Values are expressed as means ± SD or medians with interquartile range; n = 8–10 mice per group. Statistical analysis: one-way ANOVA (Newman–Keuls post hoc) and Kruskal–Wallis test (Dunn post hoc).

**Table 3 pharmaceuticals-16-00175-t003:** The effect of JJGW07 on the motor coordination in mice.

Treatment	Dose (mg/kg)	Animals That Fell from the Rotating Rod	Time before Animals Fell (s)	TD_50_ (mg/kg)
JJGW07	20	2/8	58 ± 3	28.3 (22.6–35.4)
	30	3/8	50 ± 17	
	40	7/8	20 ± 24	

Mice, previously trained for 3 consecutive days, were placed individually on a rotating rod for 60 s. The time remaining on the rod was recorded. The TD_50_ value [27] was calculated as the dose at which 50% of the animals could not stay on the rotating rod. JJGW07 was administered intraperitoneally (*ip*) 30 min before the test. Values are expressed as means ± SD; n = 8 mice per group.

**Table 4 pharmaceuticals-16-00175-t004:** The evaluation of the cataleptogenic properties of JJGW07.

Treatment	Dose (mg/kg)		Mean Score	
		30 min	60 min	120 min
	5	0.0	0.0	0.0
JJGW07	10	1.0	1.0	1.0
	20	1.3	0.4	0.1

Mice were placed on a cylindrical metal bar above the tabletop’s surface, while the hind paws remained on the tabletop. The time in which the animal held both paws on the bar was measured at 30, 60, and 120 min after an intraperitoneal (*ip*) administration of JJGW07, with a maximum measurement time of 60 s. Data are presented as the mean score for each trial, which was assessed according to Ögren et al. [28]. The minimum cataleptogenic dose was defined as the lowest dose inducing a mean catalepsy score of ≥ 1 at 30, 60, or 120 min post-treatment. n = 10 mice per group.

**Table 5 pharmaceuticals-16-00175-t005:** Radioligand binding assay conditions.

Receptor	Radioligand/Final Concentration	Blank (Non-Specific)	Buffer	Incubation Conditions
5-HT_6_	[^3^H]-LSD 2 nM	10 µM methiothepine	50 mM Tris–HCl pH 7.4 0.5 mM EDTA, 4 mM MgCl_2_	60 min, 37 °C

## Data Availability

The data are contained within the article.

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
