# Peer review of "Novel Multimodal Salicylamide Derivative with Antidepressant-like, Anxiolytic-like, Antipsychotic-like, and Anti-Amnesic Activity in Mice"

_pharmaceuticals, 2023, doi:10.3390/ph16020175_

Round 1
Reviewer 1 Report
The manuscript by Żmudzka et al., provides an important information on novel salicylamide derivate JJGW07 that as it reported has antidepressant-like, anxiolytic-like, antipsychotic-like and anti-amnesic activity in mice. The results are sounds, and the manuscript in general well written, but I have some considerations that need to be addressed before publication as follows:
1) Why does the set of drug doses vary from test to test? For example, in the tail suspension test the range 0.08 – 2.5 mg was used while in the forced swim test only 0.625 – 2.5 mg. And next, the range 0.625 – 2.5 mg was used in the FPT while the range 0.3 – 2.5 mg was used in the MBT. Authors should explain these discrepancies.
2) Object recognition test data should be presented in more informative form as discrimination or preference index (Lueptow, 2017, PMID: 28892027).
3) Authors explained the discrepancies between effects in TST and FST as a result of better predictive validity of TST (p.10, lines 283-284). This statement is absent in the original paper by Cryan, Mombereau and Vassout (2005). «Very good predictive validity» is not as the same as «better predictive validity». Authors should be careful in wording.
3) Authors interpret decrease in buried marbles as an anxiolytic effect. But in fact marble burying could be affected by any agents (e.g. benzodiazepines and 5-HT active compounds) and interventions (lesioning) affecting hippocampus function (Deacon and Rawlins, 2005, PMID: 15582110; Deacon, 2006, PMID: 17406223). Also findings by Thomas et al. (2009, PMID: 19189082) demonstrate that marble-burying/digging behavior is not correlated with exploratory activity or other measures of anxiety-like traits, as well as not affected by object novelty, nor does it change significantly with repeated testing. So, it is most likely that the JJGW07 has rather moderate anxiolytic-like properties.
Reviewer 2 Report
You generally do not use appropriately the article (the/an-a) and you should. Please revise the English of your paper. E.g., in the Abstract, you say ”Thus, this study aimed to evaluate potential antidepressant-, anxiolytic-, and antipsychotic-like effects as well as anti-amnesic properties of novel arylpiperazine derivative of salicylamide, with an affinity towards serotonin 5-HT1A, 5-HT2A, 5-HT7, and dopamine D2 receptors.” You should change it to “Thus, this study aimed to evaluate the potential antidepressant-, anxiolytic-, and antipsychotic-like effects, as well as the anti-amnesic properties of the novel arylpiperazine derivative of salicylamide, JJGW07, with an affinity towards serotonin 5-HT1A, 5-HT2A, 5-HT7, and dopamine D2 receptors”. You should name the compound here at first, not further on, when readers might wonder where this compound comes from. I underlined the additions.
Always in the Abstract, “However, pharmacotherapy of these diseases remains challenging”. These are conditions and not diseases. In psychiatry, there are no diseases, except Parkinson’s, Alzheimer’s and other dementias, whose aetiology is known. So change it to “However, pharmacotherapy of these conditions remains challenging”.
In Introduction, “central-acting compounds” should be “centrally-acting compounds”.
“In this study, we selected another arylpiperazine derivative of salicylamide, JJGW07, due to its high affinity for serotonin 5-HT1A, moderate for 5-HT7, and D2, and weak for 5-HT2A receptors [7].” This binding profile of JJGW07 does not automatically qualify it for testing in schizophrenia, depression, and anxiety models. You should specify what does it do to these receptors, whether it has an agonist, partial agonist, or antagonist function. Further on you state “Since studies show the importance of serotonin 5-HT1A and 5-HT7 receptors in depression and anxiety [9], as well as D2 receptors in schizophrenia [10,11], in this study, we aimed to assess the antidepressant-, anxiolytic-, and antipsychotic-like effect of JJGW07 in mice.” These receptors play various roles in all these conditions. For example, 5-HT1A receptors, other than being involved in anxiety (postsynaptic hippocampal 5-HT1A receptors were claimed to bring a “different way of calm” in the late 1980ies to support the use of buspirone and other azapirones, only to succumb some years later to the lack of sound clinical evidence), regulate the release of dopamine from the prefrontal cortex dopaminergic terminals (partial agonists may help negative symptoms), 5-HT7 receptors are also involved in schizophrenia and many an antipsychotic have 5-HT7 antagonist activity; furthermore, 5-HT7 inhibitors show pro-cognitive effects in memory paradigms. D2 receptors play also an important role in depression, so you cannot base your why you explore JJGW07 in the animal paradigms of depression, anxiety, schizophrenia, and cognition you chose. Since your intent was explorative, you are legitimately allowed to explore all these, but you should reformulate your phrasing. There are plenty of citations you could make in basing the above assumptions.
In Results, “JJGW07 showed no affinity for 5-HT6 receptors”, there is nothing to rejoice about; clozapine and other antipsychotics possess anti-5-HT6 receptor activity.
You should discuss this in Discussion. You should also mention among Limitations that you did not test the activity of JJGW07 on NMDA receptors, as this receptor was always involved in your tests that used dizocilpine (MK-801). You should also discuss further the -HT1A receptor antagonistic activity in the context of both antidepressant and antipsychotic activities, since it is partial agonists at these receptors that show clinically significant effects. The marble burying test is not only an anxiety paradigm, but is used also as a test for obsessive-compulsive disorder. You should comment this in Discussion.
Everything else is fine.
Round 2
Reviewer 1 Report
The authors adequately responded to all comments. I have no concernes.